# Intramuscular Polydeoxyribonucleotides in Fibrotic and Atrophic Localized Scleroderma: An Explorative Prospective Cohort Study

**DOI:** 10.3390/biomedicines11041190

**Published:** 2023-04-17

**Authors:** Maurizio Romagnuolo, Chiara Moltrasio, Angelo Valerio Marzano, Gianluca Nazzaro, Simona Muratori, Sebastiano Recalcati

**Affiliations:** 1Department of Pathophysiology and Transplantation, Università degli Studi di Milano, 20122 Milan, Italy; 2Dermatology Unit, Fondazione IRCCS Ca’ Granda Ospedale Maggiore Policlinico, 20122 Milan, Italy; 3Dermatology Unit, ASST Lecco—Ospedale A. Manzoni, 23900 Lecco, Italy

**Keywords:** localized scleroderma, Localized Scleroderma Cutaneous Assessment Tool, LoSCAT, morphea, polydeoxyribonucleotides, PDRN

## Abstract

Effective options in the quiescent, scantily inflammatory phase of localized scleroderma (morphea) are lacking. A cohort study in patients with histologically confirmed fibroatrophic morphea explored the therapeutic value of the anti-dystrophic A2A adenosine agonist polydeoxyribonucleotide (PDRN, one daily 5.625 mg/3 mL ampoule for 90 days with a three-month follow-up). Primary efficacy endpoints: Localized Scleroderma Cutaneous Assessment Tool mLoSSI and mLoSDI subscores for disease activity and damage in eighteen areas; Physicians Global Assessment for Activity (PGA-A) and Damage (PGA-D) VAS scores; skin echography. Secondary efficacy endpoints: mLoSSI, mLoSDI, PGA-A, PGA-D, and morphea areas (photographs) over time; Dermatology Life Quality Index (DLQI); skin biopsy scores and induration over time. Twenty-five patients enrolled; 20 completed the follow-up period. Highly significant improvements at the end of the 3-month treatment period: mLoSSI–73.7%, mLoSDI–43.9%, PGA-A–60.4%, PGA-D–40.3%, with further improvements at follow-up visit for all disease activity and damage indexes. Overall, the outcomes suggest that a daily PDRN ampoule intramuscularly for 90 days reduces disease activity and damage rapidly and significantly in quiescent, modestly inflammatory morphea with few currently therapeutic options. The COVID-19 pandemic and lockdowns caused difficulties in enrollment, and some patients were lost to follow-up. Due to low final enrollment, the study outcomes may have only an exploratory value, yet they appear impressive. The anti-dystrophic potential of the PDRN A2A adenosine agonist deserves further in-depth exploration.

## 1. Introduction

The dermatological entity known as localized scleroderma is a constant source of ambiguities starting with its name. Should it always be called morphea, dismissing all references to systemic sclerosis and acknowledging it is a different disease? The many similar histologic features between systemic and localized sclerosis suggest a similar pathogenesis [1]. However, the lack of fingertip ulcerations, sclerodactyly, Raynaud’s phenomenon, and autoantibodies specific to systemic sclerosis differentiate morphea from systemic scleroderma [2,3]. The unique demographics, with equal prevalence in adults and children with female gender predominance, support the differentiation [2,3]. The (uncommon) ocular, neurological, and musculoskeletal findings in morphea are also distinct from the extra-cutaneous manifestations in systemic sclerosis [3]. Raynaud’s phenomenon, antinuclear antibodies, and the typical videocapillaroscopic pictures acting as “red flags” of systemic sclerosis can help differentiate morphea from limited cutaneous systemic sclerosis, where skin lesions extend only distally from elbows and knees [4].

Unfortunately, the natural history of morphea, including how disease activity and damage evolves, has received only scant attention [5]. Adult or pediatric morphea can show a persistent disease activity or a relapsing-remitting course. Still, little is known about the time needed to suppress the disease activity with standard-of-care therapy (methotrexate, UV-A1 phototherapy), leaving doubts about whether patients have received an adequate therapy course. Even less is known about how persistent the remission of active disease is and whether damages stabilize with prolonged remission of disease activity [5]. The modest level of inflammation, even in the active morphea stages before the development of the pruriginous and sometimes painful porcelain-white or wax-yellow sclerotic plaques, complicates any judgment about the value of therapy [5,6].

The classification of the several morphea subtypes with their variable clinical expression and involvement of subcutaneous soft tissues is another source of uncertainties. Beyond the acknowledged morphea phenotypes—plaque morphea, deep morphea, bullous morphea, and generalized morphea—affecting the face with varying intensity and sometimes coexisting in the same patient, is eosinophilic fasciitis a rare morphea subtype or a distinct entity [3]? The acknowledged possibility of localized scleroderma evolving into systemic scleroderma contributes to the fog [7].

The picture typical of inflammatory activestage morphea is relapsing patches or plaques of extending pruriginous erythema and edema, with dermal and subcutaneous infiltration of lymphocytes and thickened collagen bundles [8]. The active morphea picture evolves into an inactive phase marked by hard plaques, generally starting from the center of the lesion, and eventually atrophy of the dermis and sometimes underlying soft tissues, with areas of delicate skin, hypopigmentation or hyperpigmentation, lack of inflammatory infiltrates, and loss of skin appendages [8]. Most localized scleroderma variants are less severe than pansclerotic morphea, with its circumferential involvement of limb subcutaneous tissue and muscle that may evolve into permanent joint contracture. Even in mildly or moderately severe forms, preventing the atrophic sequelae and functional impairment means early diagnosis and treatment of the hardened yellow-to-white plaques and patches with expanding violaceous borders distributed on the head, extremities, and trunk [3,9].

Current treatment recommendations include topical and systemic corticosteroids, topical calcineurin inhibitors, and systemic immunosuppressants like methotrexate, cyclosporine, or mycophenolate mofetil, all with a grievous side effect burden and all acting on the inflammatory phase. Therapeutic options targeted to the non-inflammatory fibrotic and atrophic morphea stages are currently scarce and inconclusive [1,10].

The inflammation-suppressing and anti-dystrophic effect of A2A adenosine (purinergic) receptor activation might be an innovative therapeutic strategy to control and reverse atrophy and fibrosis in quiescent morphea. Exposure to A2A agonists reduces the expression of pro-inflammatory cytokines such as Tumor Necrosis Factor-alpha and Interleukin-6, as exemplified by polydeoxyribonucleotide (PDRN), a mixture of deoxyribonucleotide polymers with chain lengths ranging between 80 and 2200 base pairs [11]. A2A agonists concomitantly increase the expression of anti-inflammatory cytokines such as Interleukin-10 [11]. The PDRN profile also includes pro-trophic effects like an increased expression of the Vascular Endothelial Growth Factor and modulation of fibroblast growth and vitality, all supporting the A2A agonist rationale against fibroatrophic progression in localized scleroderma [12,13]. Some episodic and unpublished case reports suggest a good and sometimes marked improvement of atrophy or sclerosis in morphea patients with favorable walking and functional outcomes.

The described pro-trophic PDRN profile based on the innovative A2A agonist rationale suggested performing an exploratory investigation in managing the late atrophic morphea stages, possibly leading to more demonstrative randomized trials. Thus, a preliminary probing study was designed to explore, in a small prospective cohort of quiescent morphea patients, the evolution of the fibroatrophic lesions following treatment with intramuscularly administered PDRN (Placentex^®^, 5.625 mg/3 mL ampoules, Mastelli Srl, Sanremo, Italy). Beyond its possible value in atrophic morphea management, the novel A2A agonist concept might also shed new light on how the condition arises and evolves from the earlier inflammatory stages. The PDRN-based Placentex^®^ is a prescription purinergic receptor agonist of acknowledged protrophic efficacy and safety in many disease states [13,14,15,16,17,18,19], already administered to almost two million patients before the Institutional Review Board approved the study.

## 2. Material and Methods

### 2.1. Study Design and Sample Size Estimate

Prospective, single-center, open-label cohort study with purely exploratory purpose, carried out in consecutive patients older than 18 years with quiescent localized scleroderma (morphea) confirmed histologically, with fibrotic and or atrophic cutaneous outcomes of previous actively inflammatory disease (inclusion criteria).

The study protocol, peer-reviewed and approved in 2016 by the University of Milan medical school, called for a hypothesis-probing, single-arm investigation in patients with quiescent non-inflammatory morphea, with a six-month study period—a three-month PDRN treatment period and a three-month observational follow-up. A formal sample size estimate (χ^2^ test, 0.05 two-sided significance level) had suggested an ideal sample size of 45, including drop-outs, to have an 80% power to detect a difference between the null hypothesis proportion (clinical improvement rate, 0.45) and the alternative proportion (clinical improvement rate, 0.25) [20].

The primary justification for the study’s exploratory nature was helping to design properly designed randomized studies in the future. Together with the low prevalence of morphea and quiescent morphea and related enrolment difficulties, these considerations suggested that a smaller explorative cohort of 25 morphea patients could be acceptable even if at risk of leading to falsely negative outcomes.

Exclusion criteria were steroid or systemic immunosuppressive therapy within one month before the screening, pregnancy, breastfeeding and unreliable contraception, ongoing infections in target areas, and any concomitant disorder or situation that could negatively affect compliance with study procedures (administration of at least 80% of the planned doses). The one-month washout also allowed for verifying the persistent lack of inflammatory activity in the morphea lesions. The enrolled patients had the right to withdraw from the study for any reason freely.

The investigational study center, a national reference center for localized scleroderma, should have reasonably assured the target exploratory enrolment over two years. The protocol called for including all treated patients in the Intention-To-Treat (ITT) population for statistical considerations and analysis with the Last Observation Carried Forward (LOCF) approach applied in cases of missing values. A Certificate of Clinical Trials Insurance protected the patients who signed an informed consent form. ClinicalTrials.gov Identifier in the ClinicalTrials.gov database of privately and publicly funded clinical studies conducted worldwide: NCT03388255 (accessed on 28 February 2023).

### 2.2. Treatment and Assessment Visits

Study treatment: one Placentex^®^ glass ampoule (5.625 mg/3 mL injectable solution) per day intramuscularly, corresponding to the approved daily dose (first authorization: 31 October 1994), to all patients for 90 days with tight monitoring of compliance with the prescribed study procedures. Ampoules manufactured by Mastelli Srl, the study Sponsor, according to the EU GMP (Eudralex Vol 4 Annex 13, Investigational Medicinal Products), were labeled according to EU regulations and stored at room temperature. Compliance was monitored at each visit by accounting for empty and unused ampoules. No local or systemic corticosteroids or immunosuppressive therapy was allowed over the month before the screening to avoid interferences and confirm the disease’s inactive stage.

Assessment visits: at baseline (“screening visit”) and 90 ± 7 and 180 ± 7 days after the screening visit (“end-of-treatment visit” and “follow-up visit”, respectively), without any further PRDN session after day 90.

### 2.3. Assessment Procedures

#### 2.3.1. At Each Visit

Clinical evaluation in 18 quiescent morphea areas using the validated Localized Scleroderma Cutaneous Assessment Tool (LoSCAT), articulated into an activity subscore (four-domain modified Localized Scleroderma Skin Severity Index, mLoSSI) and damage due to atrophy and fibrosis subscore (three-domain modified Localized Scleroderma Skin Damage Index, mLoSDI) [21].

Physicians Global Assessment for disease activity (PGA-A) and Damage (PGA-D) [22].

Photographs were taken at a distance of 20 to 30 cm of the patient’s fibroatrophic skin lesions to record dimensions, calculated area (mm^2^, millimeter ruler), and location.

Infrared thermography of target skin lesion to record the heat loss patterns (AGA ThermaCAM SC640, FLIR Systems, Wilsonville, OR, USA; room with temperature controlled at 22 ± 2 °C) after 15 min of acclimatization. The mean difference in temperature between the target morphea area and the contralateral healthy skin was assumed as a proxy estimate of skin sclerosis and atrophy; analysis with ThermaCAM Research Professional 2.9 software (FLIR Systems).

Ultrasound/eco-color Doppler assessment (multi-layered and single crystal technology Arietta V70 device, Hitachi Aloka Medical Ltd., Tokyo, Japan) to assess skin sclerosis and atrophy (hyperechogenicity and hypoechogenicity, respectively), the thickness of the subcutaneous adipose layer and target area blood flow.

Dermatology Life Quality Index (DLQI) self-administered, user-friendly, validated 10-item questionnaire assesses the health-related quality of life over the previous week of adult patients suffering from skin diseases [23].

Non-invasive SkinFibrometer^®^ (Delfin Technologies, Kuopio, Finland) assessment of skin induration and sub-cutaneous fibrosis, typically developing at two to four mm under the skin surface, quantitatively evaluated as the force applied on the test area when the pressure-controlled base plate is in complete contact with the indurated skin and measured with a specially designed indenter [24].

#### 2.3.2. Only at Screening and End-of-Treatment Visits

Biopsy of three pre-selected localized scleroderma areas with the scoring of the epidermal, dermal, hypodermic, and skin appendages atrophy, and dermal and hypodermic sclerosis. Serial 3-µm sections embedded in paraffin; hematoxylin/eosin stain and Dako automated cytomation immune-histochemical stain with specific CD34, clone QBEnd 10, RTU or CD31, clone JC70A, RTU monoclonal antibodies with red chromogen solution as a substrate for the enzyme.

Upon request or treatment-related toxicities, the patients removed from the study also underwent an “end-of-treatment” assessment for inclusion in the Intention-to-Treat efficacy and safety analysis.

### 2.4. Efficacy

#### 2.4.1. Primary Efficacy Endpoints

Percent of patients with clinical improvement by investigators, based on:

Composite LoSCAT scoring of, cumulatively, all inactive-stage localized scleroderma lesions (end of treatment versus baseline) with differentiation between disease activity (mLoSSI subscore) and disease damage (mLoSDI subscore); domain scores variable between zero (no disease activity or damage) and three (most severe disease activity or injury). Cumulative range of mLoSSI and mLoSDI scores in the eighteen target areas: 0 to 216.

PGA-A and PGA-D disease activity and damage scoring; end of treatment versus baseline, 10-cm visual analog scales (VAS).

#### 2.4.2. Secondary Efficacy Endpoints

Steadily assessed in a target lesion area selected at the first visit: additional LoSCAT (mLoSSI/mLoSDI) and PGA-A/PGA-D analyses and research of overall differences among all visits and post-hoc comparisons among pairs of visits to identify the points of origin in differences according to a repeated measure mixed model; dimensions of lesion areas (photographic cameras always at the same distance from lesions, assessment with a millimeter ruler); infrared thermography and echography; histologic outcomes over time (epidermal, dermal, hypodermic, and skin appendages atrophy and dermal and hypodermic sclerosis in the target skin areas; scores variable between 0 and 3); self-administered DLQI quality of life questionnaire outcome over time. The SkinFibrometer^®^ assessment of changes in skin hardening over time in target areas, especially in subcutaneous tissues, was only intended as an exploratory endpoint.

### 2.5. Safety

Throughout the study, untoward/adverse events monitoring, including any abnormal laboratory finding and worsening of a pre-existing condition by spontaneous patient reporting and active questioning at assessment visits. The reference for coding adverse events was the Medical Dictionary for Regulatory Activities Terminology (MedDRA) Version 21.0, with, as far as possible, an evaluation of the relationship to intramuscular PDRN.

### 2.6. Statistics

Descriptive statistics: means ± standard deviations, medians, and ranges for continuous variables; counts and proportions for discrete categorical variables, with the Last Observation Carried Forward (LOCF) approach, applied to missing values.

Inferential statistics for primary efficacy endpoints: preliminary conversion of LoSCAT (mLoSSI and mLoSDI) and PGA-A/PGA-D scores to categorical parameters (“Decreased”, “Unchanged”, “Increased”), then χ^2^ tests for comparing proportions between day 0 and day 90. For secondary efficacy endpoints, repeated measure mixed model or general linear model for repeated non-parametric measures (Kruskal–Wallis one-way analysis of variance on ranks) after correction for age. Post-hoc comparisons between day-90 and day-180 visits: pairwise Mann-Whitney multiple comparisons without Bonferroni correction. Confidence intervals based on Student’s approximation. For categorical parameters (“Decreased”, “Unchanged”, “Increased”): χ^2^ tests for comparing proportions between day 0 and day 90. Software (two-tailed tests): Statistical Package for the Social Sciences (SPSS, Chicago, IL, USA), version 13.0. Statistical significance for all tests: set (bidirectionally) at the threshold of 0.05.

## 3. Results

### 3.1. Efficacy

The study progressed from November 2016 until February 2019 with a lower enrolment rate than initially foreseen. By February 2019, enrolment had reached 25 patients with quiescent morphea—23 women (92%) and two men. All were of Caucasian ethnicity, between 33 and 79 years old (mean age 60.1 ± 13.0 years old, median 62), of average complexion (mean weight and height, 71.1 ± 18.8 kg, and 161 ± 8.8 cm, respectively).

Table 1 summarizes the baseline characteristics of target lesions.

The violent and disruptive explosion of the coronavirus epidemic in Northern Italy in early 2019 caused difficulties with follow-up. At that moment, 20 morphea patients had completed the study, including the final follow-up visit. Two patients out of 25 had left the study before the end-of-treatment visit (8%), increasing to five (20%), with three more patients lost before their follow-up visits because of the prolonged lockdowns and related difficulties (further details in the “Safety, drop-outs, and compliance with treatment” subchapter). In mid-2021, the authors decided to stop the paused study and analyze the available data.

At the end of the treatment, the LoSCAT and PGA subscores (mLoSSI and PGA-A Activity Indexes, and mLoSDI and PGS-D Damage Indexes), the primary efficacy endpoints, showed highly significant improvements versus baseline. This important statistical and clinical result suggests that the A2A purinergic agonist rapidly suppresses the residual disease activity in the atrophic quiescent stage and limits established and additional damage. The further improvements at day 180 compared to day 90 were statistically significant for the disease damage indexes mLoSDI and PGA-D but not significant for the mLoSSI and PGA-A activity indexes, suggesting a persistent barrier against further damage. The disease activity, already low in the morphea atrophic phase, did not improve further. Figure 1 summarizes the outcomes for the LoSCAT subscores; Figure 2 for the PGA subscores.

Other secondary efficacy endpoints also showed significant improvements in the dimension and estimated mean area of the quiescent morphea lesion (Figure 3) and the histologically assessed atrophy and sclerosis (atrophy: mean total score from 6.7 ± 2.7 at day 0 to 4.1 ± 2.3 after three months with a statistically significant mean decrease of –2.5 ± 1.92; 95% confidence interval –3.35 to –1.65; (sclerosis: mean total score from 3.9 ± 1.4 at day 0 to 2.7 ± 1.3 after three months with a statistically significant mean decrease of –1.18 ± 0.96; 95% confidence interval –1.61 to –0.76). Benefits appeared especially marked for atrophy and sclerosis in the dermis and hypodermis.

Regarding the SkinFibrometer^®^ tests, the force applied against the morphea lesions decreased after three and six months compared with the screening visit suggesting a progressive decrease of subcutaneous sclerosis measured at 2–4 mm below the skin surface (Figure 4). The mean DLQI about halved after three and six months compared with the screening visit (Figure 5), while echography imaging only showed slight, non-significant skin thickness fluctuations. Infrared thermography showed a small, non-significant, and progressively decreasing mean difference in temperature between the target lesion and the healthy control skin (from 0.98 °C measured at baseline to 0.66 °C at day 180). Figure 6 and Figure 7 (and Appendix A) show photographic examples of the outcomes of the novel anti-dystrophic PDRN strategy in scanty inflammatory, quiescent localized scleroderma with fibrotic and atrophic sequelae.

### 3.2. Safety, Drop-Outs, and Compliance with Treatment

One patient out of 25 reported a mild respiratory infection unrelated to the study treatment leading to withdrawal from the study. One other patient withdrew voluntarily for personal reasons before the 90-day treatment period was over. Three other subjects were lost to follow after the treatment period for reasons unrelated to safety problems or withdrawal of consent but because of the logistical difficulties due to the COVID-19 prolonged lockdowns.

All 25 subjects received at least one intramuscular dose of the PDRN study formulation; 23 out of 25 patients (92%) ultimately complied with the planned treatment.

## 4. Discussion

Localized scleroderma or morphea is a rare disease with an incidence of around 0.3 to 3 cases per 100,000 inhabitants/year and more common in Caucasian women, with a ratio of 2–4 women to one man. Prevalence is similar in children and adults, and the peak incidence occurs in the fifth decade in adults, while diagnosis in 90% of children is between 2 and 14 years of age [2,25,26,27].

The morphea physiopathology is still foggy, but dysfunctional fibrotic pathways certainly have a role [8]. At present, the purpose of therapy during the inactive stage of the localized scleroderma can only be to act on the connective tissue dystrophy expressing as an imbalance between deposition and degradation of collagen and leading to atrophic or sclerotic outcomes. This challenging endeavor has yet to be conclusive [1,8,10].

The literature acknowledges methotrexate, UV-A1 (340–400 nm), and psoralen UVA as the currently most effective therapies [1,28,29]. For instance, in subjects exposed to low-dose and medium-dose radiation, clinical recovery, described as fair to good response, increased from 46.2% to 72.7%, reaching 70% of considerable improvement in patients treated with high-dose UV-A1, although with a side effect burden (erythema, pruritus, burning sensations) of 15% [28]. Regarding methotrexate (e.g., 1 mg/kg per week subcutaneously), most studies report an 80% improvement rate, but the recurrence rate ranges from 28 to 44% in 16-20 months after methotrexate discontinuation of MTX [29].

The study and its validated assessment tools offer some support for the PDRN rationale in an indication and medical need with only sparse therapeutic options. In that sense, the study is innovative, with one daily 3.25-mg PDRN ampoule administered intramuscularly for 90 days improving disease activity and damage of patients with quiescent localized scleroderma. The primary endpoints, LoSCAT and Physicians Global Assessment for Disease Activity and Damage, were highly significant at day 90; the damage indexes were also at day 180.

The A2A agonist acts rapidly to extinguish the residual disease activity in the atrophic quiescent stage and erect a persistent bulwark against damage. The further improvements for mLoSDI and PGA-D over six months without favorable changes in the mLoSSI and PGA-A activity indexes suggest a long-term PDRN restructuring action that might hopefully translate into lower long-term recurrence rates. Regarding the secondary efficacy endpoints—dimension and histologic evaluations of atrophy and sclerosis of target lesion areas and experimental, objective SkinFibrometer^®^ assessment of subcutaneous induration—they also supported the hypothesis that the progressive dermal and hypodermal restructuring and anti-dystrophic actions of PDRN are veritable with a high-safety profile and a favorable impact on the health-related quality of life. The modest and evanescent changes in infrared thermography compared to healthy control skin are consistent with the non-inflammatory nature of target morphea areas.

In conclusion, the study supports the wealth of clinical experiences and preclinical and clinical pharmacology data about the adenosine A2A receptor agonist rationale in several dystrophic conditions. The study also supports the PDRN skin protrophic and restructuring effect in connective tissues. Examples are the healing of diabetic feet and torpid wounds [15,16,17] and the favorable outcomes in other challenging dystrophic disorders like lichens sclerosus [18,19].

However innovative, the study has some design liabilities. The main one is the low discriminatory power, mainly due to the difficulties during the coronavirus epidemic. Even if intended only as exploratory, the study was underpowered, much below the estimated 45 patients needed to minimize the ß-risk of false negatives. However, even a lower study enrolment appeared justified to the authors because of the forcibly limited ambitions during the difficult COVID-19 period: to give an idea of the PDRN effectiveness in atrophic morphea management and help design adequately designed and powered, randomized studies in the future. The second liability is the single-arm cohort design without an actively treated control group. Comparing PDRN with an accepted therapeutic option in atrophic morphea management would have given a more precise idea of the PDRN efficacy in the atrophic morphea severe medical need. Nevertheless, the authors believe the outcomes of their exploratory study are worthy of mention and inspiring. Of course, definitive conclusions need adequately designed and powered studies.

## 5. Conclusions

The outcomes of this innovative probing study suggest that PDRN (5.625 mg/3 mL per day intramuscularly for 90 days) rapidly and significantly reduces disease activity and damage in quiescent localized scleroderma. From its first conception, the investigation was purely exploratory with a groundbreaker, probing perspective in an indication, scanty inflammatory morphea with fibrosis and atrophy, with few effective therapeutic options. Unexpected difficulties due to the coronavirus epidemic led to unfortunate recruitment delays and the loss of some patients during lockdowns. The study outcomes are consistent across several efficacy criteria and may appear impressive, especially in a severe medical need. Despite its limitations, the study is innovative, and its value seems promising, pending further well-designed studies.

## Figures and Tables

**Figure 1 biomedicines-11-01190-f001:**
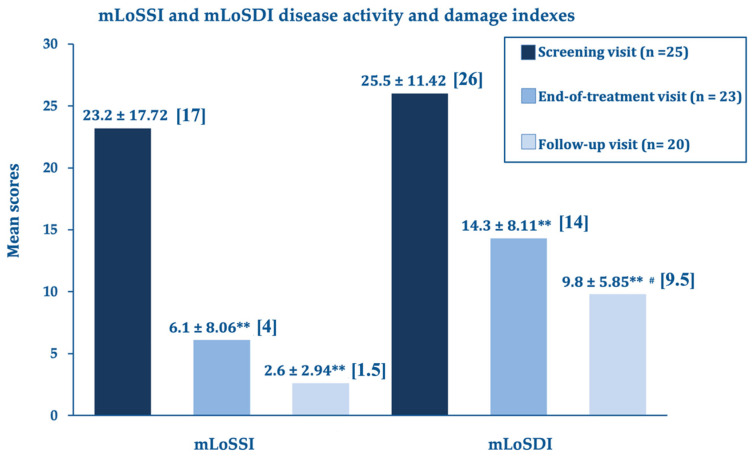
Localized Scleroderma Cutaneous Assessment (LOSCAT) subscore analysis: mean changes ± standard deviation and medians of the mLoSSI disease activity and mLoSDI disease damage indexes; end-of-treatment visit (day 90) and follow-up visit (day 180) vs. screening visit (baseline). ** *p* < 0.0001 vs. baseline screening visit, # *p* = 0.030 vs. day 90.

**Figure 2 biomedicines-11-01190-f002:**
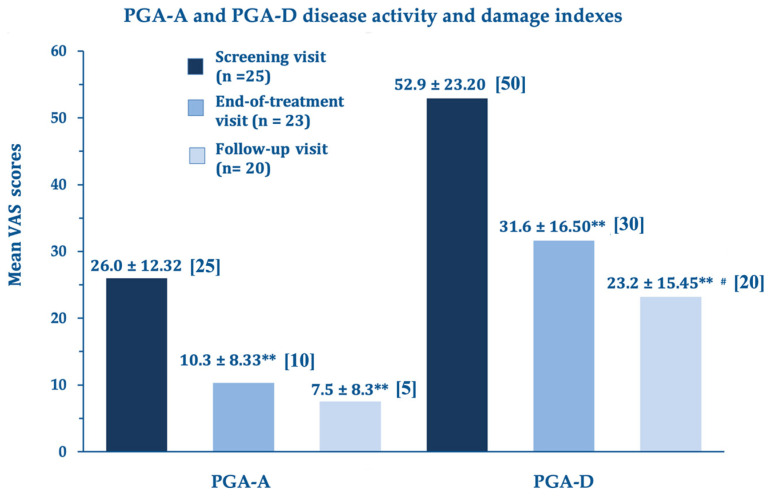
Physician Global Assessment (PGA) subscore analysis: PGA-A and PGA-D mean changes (disease activity and damage, respectively) ± standard deviation and medians, end-of-treatment visit (day 90) and follow-up visit (day 180) vs. screening visit (baseline). ** *p* < 0.0001 vs. baseline screening visit, # *p* = 0.023 vs. day 90.

**Figure 3 biomedicines-11-01190-f003:**
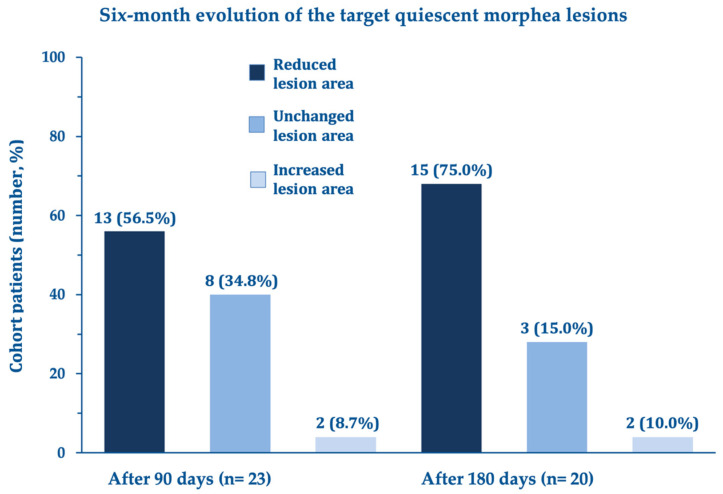
Categorical analysis of changes vs. baseline (screening visit) in the mean area of the three localized scleroderma lesions selected for biopsy after 90 (end-of-treatment visit; n = 23) and 180 days (follow-up visit; n = 20).

**Figure 4 biomedicines-11-01190-f004:**
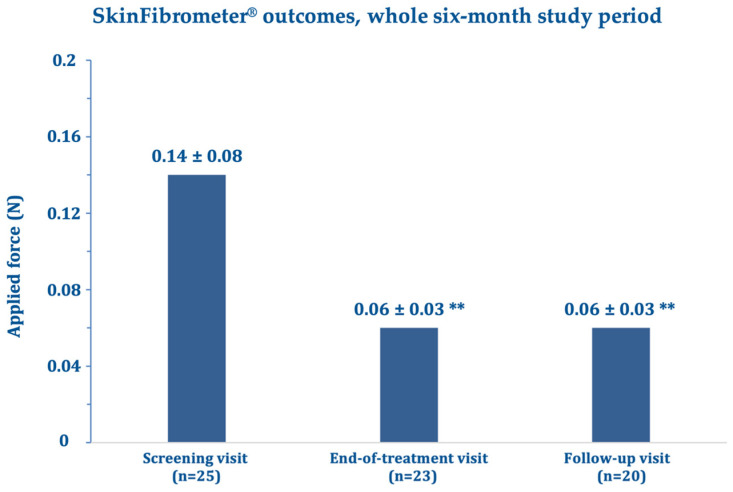
Mean changes ± standard deviation in the force applied to the target lesion area after 90 days (end-of-treatment visit) and 180 days (follow-up visit) vs. baseline (screening visit); applied forces measured in Newton (N). ** *p* <0.0001 vs. screening visit.

**Figure 5 biomedicines-11-01190-f005:**
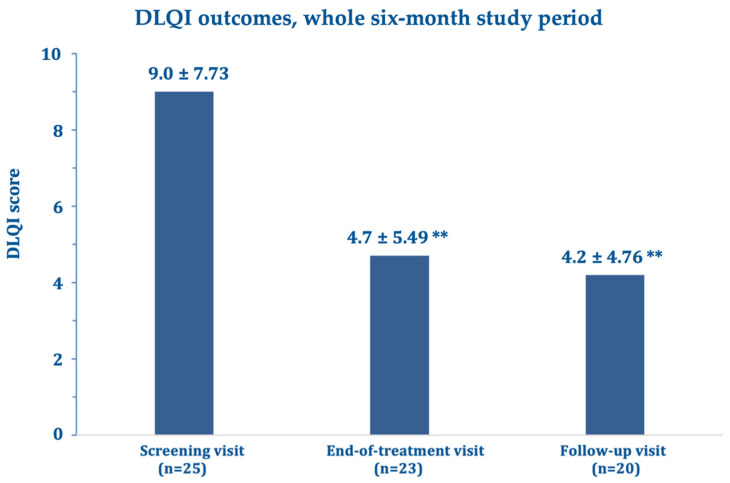
Mean Dermatology Life Quality Index (DLQI) score changes ± standard deviation after 90 (end-of-treatment visit) and 180 days (follow-up visit) vs. baseline (screening visit). ** *p* = 0.0002 vs. screening visit.

**Figure 6 biomedicines-11-01190-f006:**
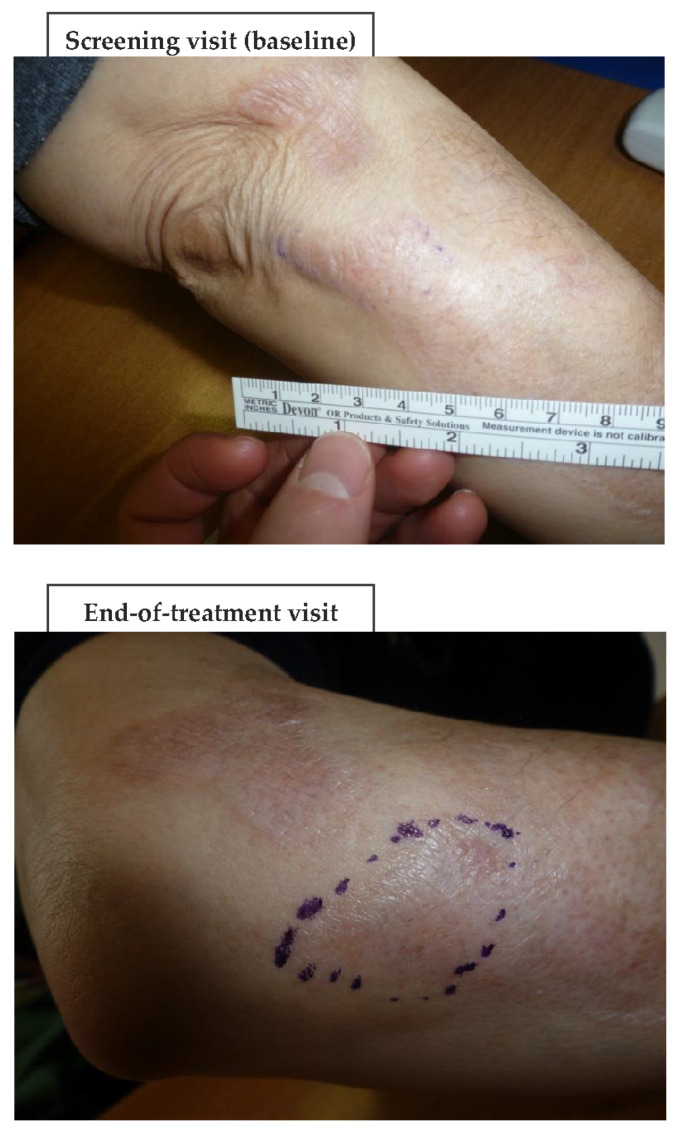
An example (progressive patient no. 6) of the benefits of a three-month PDRN treatment after 90 and 180 days (“End-of-treatment visit” and “Follow-up visit”, respectively) vs. baseline (“Screening visit”). Photographs owned by the authors with the patients’ permission of use.

**Figure 7 biomedicines-11-01190-f007:**
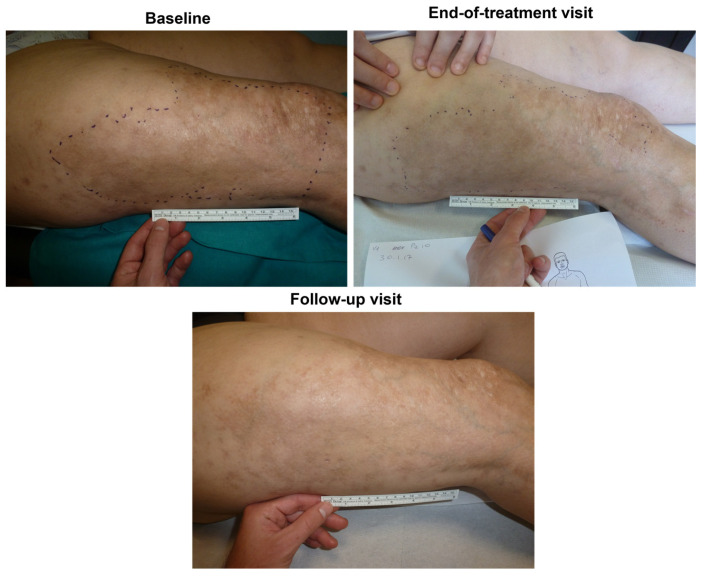
Clinical photograph of a patient (progressive patient no. 10) showing the benefits of a three-month PDRN treatment after 90 and 180 days (“End-of-treatment visit” and “Follow-up visit”, respectively) vs. baseline (“Screening visit”). Photographs owned by the authors with the patients’ permission of use.

**Table 1 biomedicines-11-01190-t001:** Quantitative description of target morphea lesions at baseline (SD: standard deviation).

Baseline Characteristics of Target Morphea Lesions	
Size of Lesions (First Diameter, mm)	
Mean ± SD	107 ± 72.8
Median	90
Min–Max	10–245
Size of lesions (second diameter, mm)	
Mean ± SD	69.3 ±47.7
Median	60
Min–Max	7–180
Temperature of lesions (infrared thermography, °C)	
Mean ± SD	31.56 ±1.66
Temperature of contralateral healthy skin (°C)	
Mean ± SD	30.86 ±1.76
Skin thickness: Atrophy	
Atrophy (ultrasound, mm): mean ± SD	2.2 ± 1.0
Sclerosis (ultrasound, mm): mean ± SD	2.2 ± 1.0
SkinFibrometer^®^ Test (force applied)	
Mean ± SD	0.1 ± 0.1

## Data Availability

The datasets generated and analyzed in the study with full details of participating patients are not publicly available. However, according to current regulations, they are archived and available from the Corresponding Author upon reasonable request after conversion in an anonymous form.

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
