# Peer review of "Intramuscular Polydeoxyribonucleotides in Fibrotic and Atrophic Localized Scleroderma: An Explorative Prospective Cohort Study"

_biomedicines, 2023, doi:10.3390/biomedicines11041190_

Round 1

Reviewer 1 Report

1) Abstract. L 28-31. The outcomes suggest  that a daily PDRN ampoule intramuscularly for 90 days reduces disease activity and damage rap- idly and significantly in quiescent scantily inflammatory localized scleroderma, a severe medical  need with few therapeutic options. Although there was a loss of patients to follow-up due to the pandemic difficulties, the study outcomes appear impressive. Please improve these paragraphs.

2) 1. Introduction L36-39 Localized scleroderma is a constant source of ambiguities starting with naming— should its name be morphea dismissing all references to systemic sclerosis and acknowl-  edging it is a different disease? Skin histologic features similar to systemic sclerosis sug-  gest that similar pathogenesis might be similar [1]. Please re-write this paragraph, it is not clear.

3) 1. Introduction. L40-42 However, the lack of fingertip ulcera- tions, sclerodactyly, Raynaud’s phenomenon, and autoantibodies specific to systemic sclerosis differentiate morphea from systemic scleroderma [2,3]. 

In order to discuss the previously described points, important references are needed to be added, such as:

a- Correlation between circulating fibrocytes and dermal thickness in limited cutaneous systemic sclerosis patients: a pilot study. Rheumatol Int. 2019 Aug;39(8):1369-1376. doi: 10.1007/s00296-019-04315-7. 

b- Autoantibodies as Biomarker and Therapeutic Target in Systemic Sclerosis. Biomedicines 202210, 2150. https://doi.org/10.3390/biomedicines10092150

c- Laser Therapy for the Treatment of Morphea: A Systematic Review of Literature. J. Clin. Med. 202110, 3409. https://doi.org/10.3390/jcm10153409

4) L 85-90. The purpose of the probing study was to explore, in a small prospective cohort of quiescent morphea patients, the evolution of the fibroatrophic lesions following treatment  with intramuscularly administered PDRN (Placentex® , 5.625 mg/3 mL ampoules, Mastelli  Srl, Sanremo, Italy). PDRN is a prescription purinergic receptor agonist of acknowledged  protrophic efficacy and safety in many conditions [10-16], already administered to almost  two million patients before the Institutional Review Board approved the study. Please, underline the novelty of the study.

5) Statistics. Please add the statistically significant value of p.

6) 3. Results. Please underline in the manuscripts the most important statisticallysignificant values.

7) 4. Discussion L298-302. Localized scleroderma or morphea is a rare disease with an incidence of around 0.3  to 3 cases per 100,000 inhabitants/year and more common in Caucasian women, with a  ratio of 2-4 women to one man. Prevalence is similar in children and adults. The peak  incidence occurs in the adults’ fifth decade of life, with 90% of children diagnosed between  2 and 14 years of age [2,22-24]. Please summarise here the most important results of the study.

8) 5. Conclusions L339-347. The outcomes of this probing study suggest that PDRN (5.625 mg/3 mL per day in-  tramuscularly for 90 days) rapidly and significantly reduces disease activity and damage  in quiescent localized scleroderma. From its first conception, the investigation was purely  exploratory; the unexpected difficulties associated with the epidemic further com-  pounded problems, with delays in recruiting patients and some patients lost to follow-up  during lockdowns. The study outcomes are consistent across several efficacy criteria. They  may appear impressive, especially in a severe medical need with few effective therapeutic  options like scanty inflammatory morphea with fibrosis and atrophy. Despite its limita-  tions, the value of this study is promising, pending further well-designed studies. Please, underline the limits of the stdy and the novelty of this work.

Reviewer 2 Report

Dear Authors,

Thank you very much for this interesting study. Due to the rarity of the disease and the difficult corona situation, it is not surprising that sufficient patient numbers were difficult to achieve.

Nevertheless, I would suggest doing everything possible to better substantiate the promising results.

Since no control group and blinding were possible, I would suggest using historical data from the clinic to describe the normal course of the disease (possibly with and without standard treatment) and/or score course over 90/180 days and, if possible, compare and discuss with the results of the current study.

In addition, of course, all available studies on fibrotic and atrophic localized scleroderma with treatment and treatment effects would have to be described in more detail, discussed and compared with the own results in introduction and discussion to mitigate the shortcoming of the missing control group and small number of patients. Since pictures are worth a thousand words, I would suggest that all available patient pictures be made available to the readership clearly anonymized as Supplementary Material so that physicians with experience can form their own opinions about the efficacy of Placentex treatment.

In Figure 6, the baseline photo is not optimal because it shows a larger section, but the interesting area shown in the other photos is somewhat cut off. Furthermore, it is not clear what the blue markings in the third photo are delimiting. Therefore, this needs to be touched up. In addition, the photos are also a bit too large and not well placed.

Round 2

Reviewer 2 Report

The authors have significantly improved their manuscript and illustrations and have adequately addressed my suggestions. Therefore, this manuscript can be accepted from my point of view.